# Pulmonary Effects of One Week of Repeated Recreational Closed-Circuit Rebreather Dives in Cold Water

**DOI:** 10.3390/medicina59010081

**Published:** 2022-12-30

**Authors:** Emmanuel Gouin, Costantino Balestra, Jeremy Orsat, Emmanuel Dugrenot, Erwan L’Her

**Affiliations:** 1TEK Diving SAS, 29200 Brest, France; 2Laboratoire ORPHY, EA 4324, Université de Bretagne Occidentale, 29200 Brest, France; 3Environmental, Occupational, Aging (Integrative) Physiology Laboratory, Haute Ecole Bruxelles-Brabant (HE2B), 1050 Brussels, Belgium; 4Anatomical Research and Clinical Studies (ARCS), Vrije Universiteit Brussels (VUB), 1090 Brussels, Belgium; 5DAN Europe Research Division (Roseto-Brussels), 1020 Brussels, Belgium; 6Physical Activity Teaching Unit, Motor Sciences Department, Université Libre de Bruxelles (ULB), 1050 Brussels, Belgium; 7Divers Alert Network, Durham, NC 27705, USA; 8Médecine Intensive et Réanimation, CHRU de la Cavale Blanche, 29200 Brest, France; 9LaTIM INSERM UMR 1101, Université de Bretagne Occidentale, 29200 Brest, France

**Keywords:** adverse effects, autonomic nervous system, decompression, lung ultrasound, mixed gas diving, pulmonary function, technical diving

## Abstract

*Background and Objectives*: The use of closed-circuit rebreathers (CCRs) in recreational diving is gaining interest. However, data regarding its physiological effects are still scarce. Immersion, cold water, hyperoxia, exercise or the equipment itself could challenge the cardiopulmonary system. The purpose of this study was to examine the impact of CCR diving on lung function and autonomous cardiac activity after a series of CCR dives in cold water. *Materials and Methods*: Eight CCR divers performed a diving trip (one week) in the Baltic Sea. Spirometry parameters, SpO_2_, and the lung ultrasonography score (LUS) associated with hydration monitoring by bioelectrical impedance were assessed at the end of the week. Heart rate variability (HRV) was recorded during the dives. *Results*: No diver declared pulmonary symptoms. The LUS increased after dives combined with a slight non-pathological decrease in SpO_2_. Spirometry was not altered, and all body water compartments were increased. Global HRV decreased during diving with a predominant increase in sympathetic tone while the parasympathetic tone decreased. All parameters returned to baseline 24 h after the last dive. *Conclusions*: The lung aeration disorders observed seem to be transient and not associated with functional spirometry alteration. The HRV dynamics highlighted physiological constraints during the dive as well as environmental-stress-related stimulation that may influence pulmonary changes. The impact of these impairments is unknown but should be taken into account, especially when considering long and repetitive CCR dives.

## 1. Introduction

The use of closed-circuit rebreathers (CCRs) has become increasingly common in the recreational scuba diving community over the past two decades. Their use allows longer and deeper dives than classical open-circuit (OC) scuba equipment. CCRs bring major advantages in terms of gas consumption, an optimal oxygen mix, and warm humidified breathing gas [1]. Conversely, since the breathing system is much more complicated to use, it exposes the diver to technical failures or specific emergencies [2].

During a dive, the cardio-pulmonary system is challenged by various combinations of stressors and adaptive mechanisms such as blood shift, thermal strain, exercise, gas density, hypercapnia, narcosis and hyperoxia [3,4,5]. In addition, the breathing apparatus by itself may add to the respiratory workload (work of breathing) that could be increased by the negative transpulmonary pressure gradient in the prone position with back-mounted counterlungs on the CCR used [6]. A number of studies have investigated pulmonary function following OC diving under varying conditions, but the results remain contradictory [4]. Many did not show any spirometric alteration after a single OC dive for a maximum depth of 65 m [7,8]. However, changes in pulmonary function have been found to be associated with depth, cold temperatures, oxidative or decompression stress and duration. These post-dive obstructive pattern changes appeared to be limited and transient [9,10]. Exposure to pure oxygen, even at shallow depths (5 msw), leads to a lung diffusing capacity alteration [11]. CCR diving exposes to high, and potentially prolonged, PpO_2_ and specific mechanical constraints [1,6]. There is a lack of data about cardio-pulmonary effects during CCR diving. A CCR deep diving study has shown an almost 30% decrease in forced vital capacity (FVC) after bounce dives at 100 msw [12]. Conversely, CCR use did not affect the spirometry despite the long duration at a maximum depth of 20 msw [13,14]. The impact of CCR repeated dives has not been evaluated in cold conditions (<10 °C) and no data are available about the evolution of such abnormalities over time. More recently, asymptomatic lung aeration defects were assessed by ultrasound with B-line detection after dives. These artefacts may suggest extravascular lung water accumulation [15,16]. Most studies have shown an accumulation of B-lines with incomplete resolution between each in repetitive deep OC dives to 60–80 msw, which is not observed at a 33 msw depth [3,8,17]. With CCR, a lung aeration loss was detected, even in shallow water, between 1 and 10 msw, and was substantially amplified by exercise and negative-pressure breathing. The right-to-left heart imbalance and increase in pulmonary vasoconstriction seem to be related to these impairments [5,6]. This phenomenon was already described during breath-hold diving and was related to diaphragmatic spasms with a closed glottis adding to the negative pressure gradient explanation [18].

All immersion constraints, such as blood centralization, pulmonary workload, or hyperoxia, also modulate the autonomic nervous system (ANS). Heart rate variability (HRV) reflects the constant fluctuation of the interaction between pulmonary ventilation, blood pressure, and cardiac output to maintain homeostasis [19]. It can be used to indirectly study changes in parasympathetic (PNS) and sympathetic (SNS) nervous system activity, which express the level of intensity in physiological adaptation. There are marked changes in autonomic cardiac activity during and after scuba diving with a predominance of PNS activity [20,21]. There are complex and sometimes conflicting additional ANS modulations, and CCR diving seems to provoke a different HRV response in divers as compared to OC diving [22].

We hypothesize that in-water breathing constraints may have a negative impact on the lung after CCR dives, especially in case of repetitive exposures. Better knowledge of the physiological impacts of CCR appears essential given the growing diving community and technical developments. Data regarding the cardio-pulmonary effects of repetitive CCR diving are still needed for different depths and environments. The aim of this study was to examine the impact of CCR diving on lung function together with autonomous cardiac activity in asymptomatic healthy volunteers after a recurrent diving exposure in cold water.

## 2. Methods

### 2.1. Diving Sites

The “Vräk diving expedition” took place in the Stockholm archipelago (Sweden) in September 2022. This study was approved by the Bio-Ethical Committee for Research and Higher Education, Brussels (No B200-2020-088), and adhered to the principles of the Declaration of Helsinki [23].

### 2.2. Study Population

This observational study was an intrasubject experimental design with repeated measures. A total of eight male divers were included. Table 1 summarizes the anthropometric data. None were smokers and none were taking medication. All of the subjects were at least recreational rebreather mixed-gas divers. The median CCR experience was 4.5 (2.9–8.3) years. Two divers (25%) have DCS history (one musculoskeletal and one lymphatic manifestations). All divers were fit to dive and had a valid medical certificate for diving. None dived in the previous week. They were all informed about the physiological study and its implications. All participants gave written consent prior to the program.

### 2.3. Diving Procedures

The dives were performed in accordance with usual CCR dive planning and were not modified for the study requirements. All dives were performed from a rigid inflatable boat. The dive sites were 30 min to 2 h away from the harbor. One to two dives were performed daily, depending on the maximal planned depth. A surface interval of 2 to 3 h between the first and second dive was met. Helium mixed-gases were used for dives below 40 m. Day 2 was taken off. The surface and bottom temperatures were 16.5 (16–17) and 8.5 (6–15.5) °C, respectively, with a thermocline at approximately 15 m. The dive parameters are shown in Table 2. Five divers performed five more comparable consecutive diving days in a second week.

Divers used back-mounted counter lung electronic controlled rebreathers (rEvo™ Rebreathers, Brugge, Belgium; *n* = 5 or JJ-CCR DiveCAN^®^, Presto, Denmark; *n* = 3). Decompression (Buhlmann ZHL-16C algorithm) was conducted using a connected Petrel 2 computer (Shearwater, Richmond, BC, Canada). The gradient factors were set to 30% (low) and 70% (high) for all dives. The oxygen partial pressure (PpO_2_) was maintained at 130 kPa during the entire dive. Subjects each wore a dry-suit with dry-gloves and an active heating system [24].

### 2.4. Measurements

Divers were monitored prior to the first dive, after the dive during the first five diving days, and 24 h after their last dive (i.e., after five diving days for three divers and ten days for five divers) in a dry and heated room (temperature 18–20 °C). The study flowchart is shown in Figure 1.

#### 2.4.1. Functional and Anatomical Pulmonary Monitoring

Measurement of pulmonary parameters was performed daily, 150 to 180 min after the dive. All measurements were performed in the sitting position and at rest. Pulse oxygen saturation (SpO_2_) and heart rate (HR) were recorded for 30 s, using a dedicated oximetry module connected to the spirometer (Spirobank II Smart; MIR Medical International Research Srl, Rome, Italy). The mean value for SpO_2_ and HR was considered. Spirometric parameters were collected including the forced vital capacity (FVC), forced expiratory volume in one second (FEV1), FEV1/FVC ratio, peak expiratory flow (PEF), and forced expiratory flow (FEF25-75) following GLI (Global Lung Initiative) 2017 for Caucasian adults [25]. The device used for measurements meets the ISO26782:2009 international standards technical characteristics and is CE marked [26]. Flow data were recorded in real time in a dedicated computer, using the manufacturer WinspiroPRO v8.1.0 software. Three repeated loops were performed to assess the reproducibility under the investigator’s supervision. The highest FVC and FEV1 values observed over the measurement series were reported [26].

The anatomical pulmonary aeration was evaluated by lung ultrasonography with a 1.1–4.7 MHz phased array probe (Venue Go™, General Electric Healthcare, Buc, France). Six areas were longitudinally scanned on each hemithorax (anterosuperior, anteroinferior, laterosuperior, lateroinferior, posterosuperior and posteroinferior) to count the total number of B-lines [27]. The exam was performed simultaneously by two trained operators to assess consistent scoring after comparison. A B-line is defined as an echogenic, coherent, wedge-shaped signal with a narrow origin arising from the hyperechogenic pleural line and extending to the far edge of the viewing area [15]. The amount of lung aeration loss was calculated in a semi-quantitative approach using the validated lung ultrasound score (LUS). For each explored region, the worst finding was reported according to the following rating: normal: 0; well-separated B-lines: 1; coalescent B-lines: 2; and consolidation: 3 [27]. The LUS corresponded to the sum of each scanning site score (range 0 to 36). An increase in score indicates a decrease in lung aeration without necessarily reaching pathology levels [28].

#### 2.4.2. Hydration Status

All divers had unrestricted access to drinking water. Bioelectrical impedance analysis (BIA) is a safe and fast method to evaluate the body composition or hydration status [29,30,31]. These changes were estimated by a multifrequency tetrapolar impedancemetry Biody XPert^ZM^ (Aminogram, La Ciotat, France) in order to evaluate the total body hydration status and its related changes after diving. It is presented as a hand–foot analyzer and the measuring time is within the minute. Data were measured according to the manufacturer’s instructions in a seated position before the first dive, 150 to 180 min after the first-day dive and at the end of the fifth-day dive, and 24 h after the last-day dive. The data were directly transferred via Bluetooth using the proprietary Aminogram Biodymanager app for Android. The device is accredited to the ISO13485:2016 standard and is CE marked. The response of different body tissues to the application of a weak alternating current at five different frequencies (range 5 to 200 KHz) determines the resistant indices (IRs) and the phase angle at 50 kHz (PA°). It allows an estimate of the total body (TBW), intracellular (ICW) and extracellular (ECW) water. The phase angle expresses both changes in the amount as well as the quality of soft tissue mass (i.e., cell membrane permeability and soft tissue hydration) [32,33].

#### 2.4.3. Heart Rate Variability

Heart rate variability (HRV) is a non-invasive assessment of the variation in time between consecutive inter-beat intervals (R–R intervals) that results in a dynamic relationship between PNS and SNS [19]. It represents the ability of the heart to respond to a variety of physiological and environmental stimuli. Each diver wore a chest elastic belt sensor Polar H10 connected to the Polar Unite watch (Polar Electro Oy, Kempele, Finland) to record the R-to-R interval at a 1000 Hz sampling frequency. The validity of this device for HRV measurement has already been demonstrated in several studies [22,34]. The resting baseline was recorded in a sitting position, after ten minutes at rest, before the first dive and 24 h after the last day for ten minutes. During diving, the Polar watches were placed on the shoulder strap of the undergarment inside the dry suit after starting recording. Full data were extracted daily and analyzed using the Kubios HRV Premium Analysis Software 3.5.0 (UKU, Kuopio, Finland). The automatic artifact correction function of the program was used to correct data corruption for each subject before analysis. Time–domain results with mean HR, standard deviation of normal-to-normal R waves (SDNN), root mean square of the successive difference (RMSSD) of the R–R intervals, and integral of the density of the R–R interval histogram divided by the maximum of its weight (RR triangular index) were calculated. RMSSD mainly reflects the parasympathetic tone while the SDNN and triangular index are indicators of the overall ANS activity frequency-domain measures including the very-low-frequency (VLF), low-frequency (LF), and high-frequency (HF) spectral absolute power and LF/HF ratio. These estimate the distribution of absolute or relative power in different frequency bands [35]. Quantitative analyses of Poincare plot features (SD1, SD2, and SD2/SD1 ratio), Shannon entropy (ShanEn), and Multi-scale entropy (MSE) were computed. This non-linear approach is less dependent on the respiratory sinusal arrythmia variations in the R–R intervals [22]. Furthermore, two composite indexes were calculated. These indexes are based on known HRV parameters that reflect PNS and SNS activity. PNS and SNS indexes were based on the mean R-to-R interval, RMSSD and SD1, and the mean HR interval, the stress index, and SD2, respectively [36]. An index value of zero reflected that the PNS or SNS activity is equal to the normal population average [37].

## 3. Statistical Analysis

Statistical analysis was performed with GraphPad Prism v9.0.2 (GraphPad Software Inc., San Diego, CA, USA). All data are presented as the median (first and third quartile). The normality of distribution was assessed by Shapiro–Wilk test. ANOVA for repeated measures was used to analyze more than two related groups followed by multiple comparison Tukey’s post hoc test. A two-way ANOVA for repeated measures was used to assess the effects of the dive day and chest site on the LUS. If non-normality was found, a non-parametric Friedman test was used followed by multiple comparison Dunn test. Statistical significance was set at a *p*-value < 0.05.

The sample size was calculated setting the power of the study at 95% and assuming that variables associated with diving would have been affected to a similar extent as observed in our previous studies where our sample reached 98% [12,30].

## 4. Results

### 4.1. Respiratory and Pulmonary Parameters

Divers performed the repetitive diving program without any pulmonary symptoms or any other disturbance. No modification of spirometry parameters was observed (Table 3). The mean SpO_2_ significatively decreased during all diving days by −1.4 [−2.6; −0.8] % (F = 5.02, *p* = 0.02). Few B-lines were observed on the baseline for several divers, with a median LUS at 1.5 [0.3; 2.8] and a basal predominance.

There was significative B-line accumulation after diving days 3, 4, 5 and 6 for all participants (Figure 2, F = 8.50, *p* = 0.003) with a return to baseline 24 h after the last dive (*p* > 0.99). There was no difference in chest site repartition (F = 1.02, *p* = 0.4) between lung territories or interaction with the day effect (F = 0.67, *p* = 0.9).

### 4.2. Impedancemetry

All body water compartments and TBW increased after dives. Figure 3 depicts variation as compared to the baseline. No significative variation was found 24 h after dives in each compartment. A trend towards an IR (impedance ratio) decrease between baseline and after dives was observed at D1 and D6 (−1.5 [−2.3; −0.2] % and −1.85 [−3.4; −0.6] % F = 4.35, *p* = 0.06, respectively). There was no change in the angle phase (F = 0.20, *p* = 0.8).

### 4.3. Heart Rate Variability

Only 23 measurements of HRV were accurately recorded during dives due to methodological issues. HRV data are shown in Table 4. No change was observed between baseline and 24 h post-dive. RMSSD and HF were significatively lower in dive versus rest measurements. Similarly, the global activity showed a decrease in variability with lower triangular index, SD2/SD1 and LF/HF ratios. The SDNN decrease was not significative (F = 5.83, *p* = 0.05).

At baseline, the PNS index was −1.92 [−2.17; −0.69] and decreased to −3.26 [−2.17; −2.8] on immersion (*p* < 0.001). Conversely, the SNS index varied to 5.570 [4.97; 6.62] at 13.5 [9.38; 19.72] (*p* = 0.02). The PNS index decreased on immersion (F = 35.83, *p* < 0.0001) while the SNS increased (F = 23.74, *p* < 0.0001), as shown in Figure 4.

## 5. Discussion

The present study depicts that lung aeration disorders are observed during repeated CCR dives. However, these abnormalities, associated with a slight but significant SpO_2_ decrease, may be transient and not associated with lung spirometry modifications.

A long-term FVC increase has already been reported in the experienced diving community, in relation to the chest muscular workload at depth. This change suggests a distension of the alveoli wall that may cause narrowing of small airways [4]. It could explain the moderate but not significant basal FEF 25–75 alteration observed in our study. However, the absence of any significant spirometric alteration after such shallow dives is similar to previous studies on CCR diving. There is a general consensus in pulmonary medicine and anesthesiology that breathing oxygen at an oxygen partial pressure (PpO_2_) higher than 50 kPa causes acute pulmonary injury, which can result in atelectasis, interstitial oedema, and inflammation [38]. In such diving conditions, there was no significant clinically relevant impairment of clinical airway physiology. After breathing PpO_2_ at 140 kPa for 20 min at 15 msw, an increase in oxidative stress urinary markers has been described but was not considered sufficient to affect the spirometry [14]. The pulmonary function also remained unchanged either after a prolonged 3- and 12 h exposure at 5 msw and 20 msw, respectively, with a PpO_2_ of 150 kPa or after repeated dives (20 dives within 11 days) at an average depth of 69 msw during 112 min with a PpO_2_ set at 130–140 kPa. It should be noted that the recommended maximal repetition excursion oxygen exposure (REPEX) threshold was approached in these studies [11,13,39]. In contrast, for deeper dives (90–120 msw) with a duration of 2 or 3 h, we previously found a gradual FVC decrease from 109 to 73% of the predicted value after a second dive, without returning to baseline between dives. No alteration of pulmonary resistance was observed, which might suggest other physiological mechanisms than hyperoxia. Considering all these arguments, one might consider that the alteration of spirometry data seems more likely to result from the effects of prolonged and deeper immersion at depth than from oxygen toxicity by itself [12].

A loss of lung aeration was observed after dives, as shown by the accumulation of B-lines without the alteration of spirometry. B-lines are an index of extravascular congestive lung fluid, which has been previously validated with high sensitivity and intra-patient reliability, allowing good interrater consistency of pulmonary fluid assessment using radiographic imaging [16,28]. Some authors have reported up to 75% of divers showing extravascular lung water detected as B-line accumulation. Many factors seem to be associated with asymptomatic changes in cardiovascular and pulmonary physiology in diving, therefore linked to the development of extravascular lung water [40]. However, B-lines are not specific, and their occurrence may also reflect any interstitial disorder or ventilation impairment. Some studies indicate a good correlation between their number and the intensity of damages [41]. An aeronautic study has shown that hyperoxia and hypergravity are independent risk factors of pulmonary atelectasis formation in healthy humans after a long arm centrifuge session. The increase in B-lines has been reported to reflect the onset of hyperoxic atelectasis [42]. Our study does not enable us to distinguish extravascular lung water or atelectasis contribution. It is interesting to note, while not significant, that a higher LUS number was observed during the first two days. Dives were shallower but the total immersion time and oxygen exposure were longer. Moreover, helium mixed gases were used for the deepest following dives, thus inducing a lesser gas density and a decrease in breathing workload. Two or three hours after surfacing from a deep Trimix dive, the B-lines were already largely resolved, similar to what has been reported by a previous Croatian study with similar dives [8]. Our results suggested that most of the pulmonary changes including loss of aeration lasted only for a short time after dives with a return to baseline 24 h post-dive.

A reduction in pulmonary diffusing capacity was shown only after a wet shallow oxygen dive as compared to a dry similar dive that suggested the implication of cardiopulmonary changes in immersion [11]. This impairment was inconsistent and was not correlated with the presence of B-lines [8]. In our study, SpO_2_ slightly decreased after dives but remained within physiological values considered to be normal [43]. This oxygenation decrease may be related to the lung aeration loss and a potential alteration of alveolo-capillary gas exchange, even though it persists while LUS values have decreased and/or returned to baseline values. Atelectasis could lead to a pulmonary ventilation/perfusion mismatch and shunt opening [42]. Although non-pathological, this may interfere with gas elimination and decompression. Similar results were found after CCR dives at 10 msw [5] but not after deep dives despite spirometric alteration [12]. This SpO_2_ decrease might be compensated by a prolonged high PpO_2_ up to 150 kPa during long decompression stop after deep dives. Several artefacts can interfere with SpO_2_ monitoring. However, data were always recorded after rewarming, in dry conditions, and after hydration in order to reduce these methodological artefacts [43].

It is well known that immersion induces hyperdiuresis, which in turn alters the hydration status with a loss of body weight of up to 3% and a potential impact on the cardio-pulmonary system [12,44]. Conversely, our results showed an increase in body water after diving. Considering that dehydration plays a role in decompression stress and that water intake could provide a decrease in the risk of decompression sickness (DCS), no specific instruction was given to fluid management for the team [45,46]. Technical divers were aware of this problem, and they probably rehydrated themself effectively during hours prior to measurements. There is no direct evidence within the literature that immersion pulmonary oedema is related to hydration in healthy divers [47,48]. In our study, there is no clear evidence that the observed state of hyperhydration could have contributed to lung ultrasound abnormalities.

HRV monitoring during scuba diving is only available from a limited number of studies, and CCR diving seems to induce a different HRV response than OC diving [22,49]. Immersion stimulates both the PNS and SNS branches. A predominant PNS is usually observed during descent and bottom stay, in accordance with human diving responses [49]. After emersion and continuation of atmospheric air breathing, the SNS takes over the PNS. A PNS tone increase in dive, related to dive length and depth, has been demonstrated [50]. Nitrox diving seems to induce a higher PNS activation [51] but also to be the principal dynamic component of SNS [50]. Short-term HRV is in fact influenced by many other factors, including PNS/SNS balance, as well as respiration via the respiratory sinus arrhythmia, heart and vascular tone via baroreceptor and cardiac stretch receptor activity, the central nervous system (CNS), the endocrine system, and chemoreceptors [21]. The PNS activity is a major contributor to the HF component (which reflects the power of vagally modulated respiratory sinus arrhythmia) [19]. Non-linear analysis revealed complexity in heart-rate patterns, which could not be perceived from time–domain [52]. Compared to OC diving, no variation in HF power or SDNN at depth was found in CCR diving, while the non-linear analysis increased. This suggests a lower PNS dynamic and variability in CCR dives [22]. In CCR cold diving (2 to 4 °C), the PNS index significatively decreased at submersion and increased gradually throughout the dive. At the same time, the SNS index sharply increased during immersion and then slowly declined back to the pre-dive rest levels [21]. Our results demonstrate a similar PNS dynamic. However, the SNS index appeared mostly predominant while global HRV activity and PNS decreased, which is contradictory to previous observations during OC diving. The SNS activity can be stimulated by many factors including physical activity or psychologically stressful situations [53]. The divers wore heavy equipment and were not necessarily previously familiar with the diving conditions that they experienced in this dynamic. Our HRV analysis has some limitations as we were only able to record 23 measurements due to interference between the heating system and the sensor. Moreover, the dive profiles varied over time, and we are not able to compare the responses at these different depths.

## 6. Limitations

The findings of this study have to be viewed with caution due to the small number of subjects and lack of daily pre-dive measurements due to logistical constraints. All measurements are compared to the first-day dive baseline, which reduces the accuracy in the assessment of physiological variations throughout the program. That may have particularly affected our impedancemetry analyses. In addition, the lack of monitoring fluid intake makes the interpretation difficult.

The mandatory boat travel-back time led to many hours of delay before measurements. Thus, we are aware of the risk of missing potential transient changes in the measured parameters. Our study was not conducted in controlled laboratory-like conditions. Essential parameters that could interfere with our results such as temperature, visibility, current, depth, or dive duration could not be controlled. Unfortunately, due to the small number of dives, the effect of the dive profile and breathing mixture cannot be evaluated.

## 7. Conclusions

The present observation represents the first original data regarding the pulmonary effects of repetitive CCR dives, combining spirometry, oxygenation evaluation and lung ultrasound imaging. Despite no detectable change in pulmonary function, we observed a significative loss of lung aeration. The impact of these impairments is unknown but should be taken into account, especially when considering long and repetitive dives. The marked changes that were also observed in autonomic cardiac activity highlight the important physiological and environmental constraints in CCR diving. All cardiac and pulmonary function changes were, however, transient, without the negative effect of dive repetition, and returned to baseline within 24 h after the last dive. Further research on this topic is encouraged to gain better knowledge about cardiopulmonary constraints during CCR diving.

## Figures and Tables

**Figure 1 medicina-59-00081-f001:**
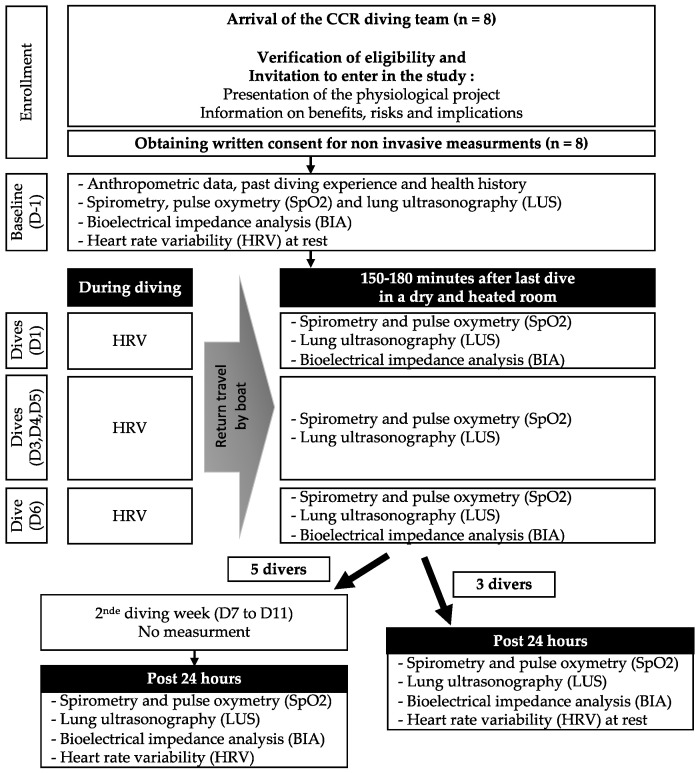
Experimental flowchart.

**Figure 2 medicina-59-00081-f002:**
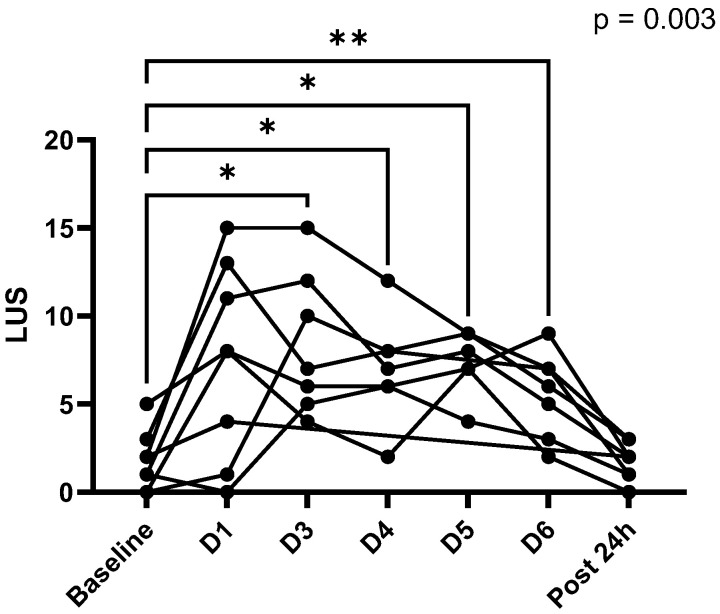
Evolution of individual lung ultrasound score (LUS) throughout repetition of day dives. All divers developed B-lines after several dive days and a return to baseline 24 h after the last one. *p*-value for the ANOVA. Tukey’s multiple comparisons test is expressed versus baseline with * *p* < 0.05, ** *p* < 0.01.

**Figure 3 medicina-59-00081-f003:**
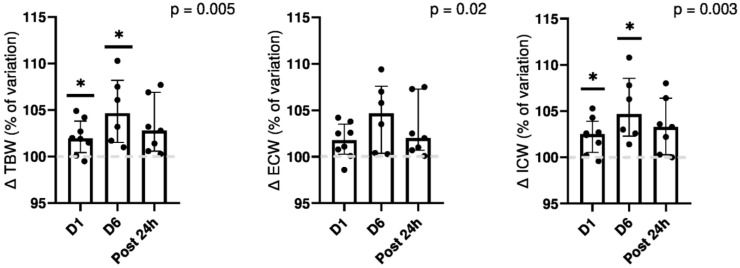
Evolution of hydration status measured by impedancemetry. Results are expressed in percentage of baseline variation. TBW, total body water; ECW, extra-cellular water; ICW, intra-cellular water. The *p*-value for the ANOVA. Tukey’s multiple comparisons test is expressed versus baseline with * *p* < 0.05.

**Figure 4 medicina-59-00081-f004:**
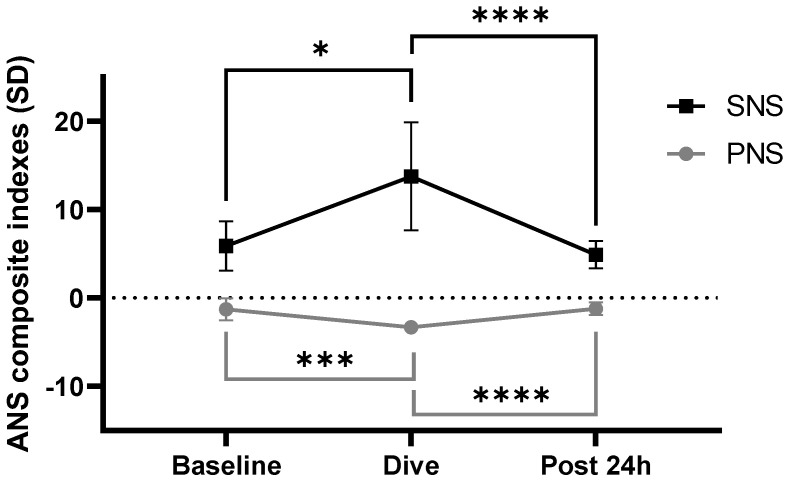
Autonomic nervous system (ANS) composite index measured by HRV. HRV parameters were performed at rest, in sitting position at baseline and 24 h after last day dive. Dive measurements were recorded in immersion during the diving program (*n* = 23). PNS, in parasympathetic nervous system; SNS, sympathetic nervous system; SD, standard deviation of measurement to the normal population average. Dunn’s multiple comparisons test is expressed versus dive with * *p* < 0.05, *** *p* < 0.001 and **** *p* < 0.0001.

**Table 1 medicina-59-00081-t001:** Participants’ anthropometric parameters (*n* = 8).

	Median (1st–3rd Quartile)
Age (years)	40.5 [36.25; 47]
Weight (kg)	80 [75; 103]
Height (cm)	178 [175; 187]
BMI (kg.cm^−2^)	25.7 [23; 30.3]

BMI, body mass index.

**Table 2 medicina-59-00081-t002:** Diving description during the monitoring week. The third dive on the first day were performed by 5 divers to test equipment. Cumulative central nervous system oxygen toxicity (CNS) represents the time spent at a given oxygen partial pressure (PpO_2_) and dividing by the NOAA time limit for that PpO_2_ (corresponding to the cumulative oxygen exposition).

	D1 (2 or 3 Dives) Air Diluent	D3 (2 Dives) Air Diluent	D4 (2 Dives) Air Diluent	D5 (1 Dive) Trimix 15/40	D6 (1 Dive) Trimix 10/50
Maximal depth (msw)	19.7 [13; 20.9]	29.3 [28.6; 33.8]	32.9 [28.6; 37.8]	44.8 [41.6; 44.9]	64.7 [46.1; 70.1]
Mean depth (msw)	14.3 [9.3; 16.0]	22.3 [20.8; 23.3]	22.0 [21; 22.3]	24.2 [23; 24.5]	24.1 [23.6; 24.3]
Each dive time (min)	54 [41; 63]	66 [55; 94]	64 [58; 73]	99 [83; 99]	90 [88; 92]
Total dive time (min)	147 [98; 172]	136 [93; 168]	126 [109; 141]	99 [83; 99]	90 [88; 92]
Cumulative CNS (%)	38 [22; 48]	47 [39; 58]	43 [34; 50]	52 [46; 53]	49 [45; 52]

**Table 3 medicina-59-00081-t003:** Spirometry parameters. Data were assessed at baseline and 150–180 min after each day dive. FCV, forced vital capacity; FEV1, forced expiratory volume in one second; PEF, peak expiratory flow; FEF2575, forced expiratory flow; SpO_2_, pulse oxygen saturation; HR, heart rate. Data are expressed in absolute value and the percentage of expected values according to the GLI (% pred). *p*-value for the ANOVA. Tukey’s multiple comparisons test is expressed versus baseline with * *p* < 0.05. Differences versus post-24 h are expressed with ^†^
*p* < 0.05 and ^††^
*p* < 0.01.

	Baseline	D1	D3	D4	D5	D6	Post 24 h	*p*-Value
FVC (l) (% pred)	5.6 [4.8; 6] 106 [101; 111]	5.8 [5.1; 6.5] 112 [100; 117]	6.0 [5.9; 6.6] 115 [108; 123]	5.9 [5.4; 6.5] 112 [106; 118]	6.2 [4.9; 6.5] 107 [101; 115]	6.2 [5.4; 6.4] 112 [102; 119]	6.1 [5.2; 6.4] 107 [104; 117]	0.2
FEV1 (l) (% pred)	4.4 [4.0; 4.5] 101 [96; 111]	4.5 [4.2; 4.7] 104 [98; 117]	4.5 [4.1; 4.7] 107 [94; 113]	4.6 [4.3; 4.7] 103 [95; 116]	4.6 [4.0; 4.8] 100 [93; 113]	4.5 [4.2; 4.8] 101 [91; 114]	4.5 [4.3; 4.6] 105 [95; 113]	0.5
FEV1/FVC (%) (% pred)	78 [72; 84] 96 [89; 105]	78 [71; 81] 97 [88; 101]	74 [70; 78] 91 [88; 97]	78 [72; 79] 97 [89; 99]	77 [72; 81] 95 [89; 102]	77 [72; 78] 94 [89; 97]	77 [71; 82] 96 [88; 102]	0.4
PEF (l.s^−1^) (% pred)	11.6 [11.1; 12.2] 125 [117; 134]	12.3 [10.9; 13.0] 133 [115; 138]	11.7 [10.7; 12.4] 122 [115; 136]	12.0 [10.9; 12.5] 132 [121; 134]	11.6 [10.4; 12.6] 120 [110; 129]	11.7 [10.6; 13.2] 123 [108; 132]	11.5 [10.6; 13.6] 125 [113; 147]	0.4
FEF2575 (l.s^−1^) (% pred)	3.8 [3.1; 4.7] 95 [74; 120]	3.7 [3.0; 4.9] 97 [75; 116]	3.3 [3.1; 3.6] 82 [70; 97]	3.4 [3.3; 4.0] 92 [71; 113]	3.8 [3.2; 4.4] 86 [74; 115]	3.4 [3.0; 4.2] 97 [67; 100]	3.5 [3.2; 4.5] 91 [72; 119]	0.8
SpO_2_ (%)	97.8 [97.4; 98.4]	96.2 [95.7; 97.2] *	96.6 [95.5; 97.3] *	96.5 [96.3; 97.1]	96.6 [95.5; 97.1]	96.2 [95; 96.7] *	97.4 [96.7; 97.7]	0.02
HR (bpm)	78 [67; 86]	98 [97; 104] *^,††^	94 [87; 103] *^,†^	94 [84; 99] ^†^	98 [96; 104] *^,†^	91 [80; 97]	80 [71; 83]	<0.0001

**Table 4 medicina-59-00081-t004:** HRV parameters. Data were recorded at rest, in sitting position at baseline and 24 h after last day dive. Dive measurements were recorded in immersion during the diving program (*n* = 23). HR, heart rate; SDNN, standard deviation of normal-to-normal R waves; RMSSD, root mean square of the successive difference; RR triangular index, R–R intervals, and integral of the density of the R–R interval histogram divided by the maximum of its weight; VLF, very low-frequency; LF, low frequency; HF, high frequency; SD1, beat-to-beat HR variability; SD2, global HR variability; ShanEn, Shannon entropy; MSE, multi-scale entropy. *p*-value for the Friedman test. Dunn’s multiple comparisons test is expressed versus dive with * *p* < 0.05, ** *p* < 0.01, *** *p* < 0.001 and **** *p* < 0.0001. Differences versus baseline are expressed with ^†^
*p* < 0.05 and ^††^
*p* < 0.01.

	Baseline (Rest)	Dive	Post 24 h (Rest)	*p*-Value
Time domain				
Mean HR (bpm)	90 [65; 92] ***	118 [108; 135]	75 [74; 82] ****	<0.0001
SDNN (ms)	54.6 [51.4; 95]	35.7 [28.2; 65.3]	53.9 [53; 81.8]	0.05
RMSSD (ms)	24.2 [20.3; 26.6] **	6.1 [3.3; 9.1]	20.4 [19.8; 38.9] ****	<0.0001
RR triangular index	14.47 [13.21; 18.25] *	9.71 [7.99; 15.12]	14.41 [11.07; 22.05]	0.02
Frequency domain				
VLF (ms2)	1302 [947; 2683]	802 [467; 1908]	1843 [1780; 1932]	0.1
LF (ms2)	1259 [1000; 2445] ****	69 [28; 210]	775 [773; 2885] **	<0.0001
HF (ms2)	117 [92; 253] **	10 [3; 50]	134 [97; 629] ****	<0.0001
LF/HF ratio	11.35 [4.936; 13.68] *	5.764 [3.629; 9.095]	5.75 [4.585; 7.504] ^†^	0.01
Non-linear results				
SD1 (ms)	17.1 [14.4; 18.8] ***	4.3 [2.3; 6.4]	14.4 [14; 27.5] ***	<0.0001
SD2 (ms)	74.8 [71.3; 133.2]	50.1 [39.8; 78.3) ]	74.9 [73.4; 112.4]	0.05
SD2/SD1 ratio	4.965 [3.976; 5.553] ****	12.59 [7.818; 18.04]	5.187 [4.087; 5.249] ****	<0.0001
ShanEn	3.37 [2.946; 3.47] ****	4.238 [3.934; 4.553]	3.463 [3.102; 3.578] ***	<0.0001
MSE min	0.758 [0.711; 0.892]	0.658 [0.447; 0.797]	1.044 [0.914; 1.124] ****^, ††^	<0.0001
MSE max	2.787 [2.224; 2.902] ****	1.544 [1.358; 1.79]	2.485 [2.208; 2.65] ***	<0.0001

## Data Availability

Data are available on request from the authors.

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
