# Peer review of "Pulmonary Effects of One Week of Repeated Recreational Closed-Circuit Rebreather Dives in Cold Water"

_medicina, 2022, doi:10.3390/medicina59010081_

Round 1

Reviewer 1 Report

I suggest acceptance following some corrections

Major

This study contribute to the safety in multiple dives using CCR and worth publication. I do not have major critics.

Minor

Table 1 It seems to me better to divide it to two tables.

Rephrase in the title "diving characteristic" for better understanding.

Define what is in the brackets once 3 numbers and in other two. This is applicable for the next table as well.

In table 2 you define the different vday by D. Replace J for D in this table.

In the same day one dive may affect the following dive. It would be nice to add the time interval between sequence of dives.

What do you mean by "End day CNS" Is it a risk if so what was the algorithm.

Table 2 "Results are in bold" I could not observe bold in significant change. The different symbols make it clear

Next Table 2 Should be 3 or either 4 if you separate tabl1 1 to 2.

Author Response

Thanks a lot for your careful revision

All comments have been taken into account

Tables have been modified accordingly

See modifications within text for other comments, all being highlighted i red

Specific answers:

Table 1 It seems to me better to divide it in two tables.

  • The table has been divided accordingly

Rephrase in the title "diving characteristic" for better understanding.

  • Characteristic has been changed into “description” for clarification

Define what is in the brackets once 3 numbers and in other two. This is applicable for the next table as well.

  • Corrected

In table 2 you define the different vday by D. Replace J for D in this table.

  • Done accordingly – thank-you!

In the same day one dive may affect the following dive. It would be nice to add the time interval between sequence of dives.

  • We added some precisions in the manuscript: surface interval 2-3 hours.

What do you mean by "End day CNS" Is it a risk if so what was the algorithm.

  • End day CNS has been changed into -> cumulative CNS (%) some more explanation added.

Table 2 "Results are in bold" I could not observe bold in significant change. The different symbols make it clear

  • The sentence was confusing we therefore delete it and clarified.

Next Table 2 Should be 3 or either 4 if you separate tabl1 1 to 2.

  • Done accordingly

Reviewer 2 Report

medicina-2084361

Pulmonary effects of one week of repeated recreational Closed-Circuit Rebreather dives in cold water. 

Summary:

medicina-2084361 “Pulmonary effects of one week of repeated recreational Closed-Circuit Rebreather dives in cold water” is a well-presented study of

Overall comments:

This relatively simple study demonstrating that there are little to no pulmonary or autonomic system effects with repeated closed-circuit rebreather use during scuba.  There are significant limitations mentioned at the conclusion of the manuscript that might be brought up earlier during the methods section.  There are significant English translation issues that need to be rectified.

Specific comments

There are frequent English language use issues, for example,

Abstract: Results: Typographical error, missing s: “No diver declared pulmonary symptom.” should be “No diver declared pulmonary symptoms.”

Page 2, Paragraph 1, Line 5: Typographical error: “exposes more frequently” should be more frequently exposes.

Page 2, Paragraph 2, Line 3: Typographical error: “In addition, breathing apparatus” should be “In addition, the breathing apparatus . . .”

The study purpose is stated at the end of the introduction.  Did the authors generate a hypothesis for the study?

The mean number of years diving on CCR is provided for the participants, can the mean number of dives on CCR also be provided as this provides more meaningful detail for the divers’ actual experience levels.

Figures and Tables:

Table 1.  The bottom portion of the table is not clear.  Can this be more fully explained, and all of the abbreviations defined (should the J’s across the top be D’s for day?)?

Figure 1.  There are word spacing issues within the text boxes.

References

No comments.

Author Response

Thanks a lot for your revision

All comments have been taken into account

Tables have been modified accordingly

An english revision has been performed (see certificate)

Specific comments:

There are significant limitations mentioned at the conclusion of the manuscript that might be brought up earlier during the methods section.  

Limitations have been updated

There are significant English translation issues that need to be rectified.

The manuscript has been proofread and by an expert and every issue has been addressed.

The study purpose is stated at the end of the introduction.  Did the authors generate a hypothesis for the study?

  • This sentence has been added for clarification: We hypothesis that in-water breathing constraints may have a negative impact on the lung after CCR dives, especially in case of repetitive exposures.

The mean number of years diving on CCR is provided for the participants, can the mean number of dives on CCR also be provided as this provides more meaningful detail for the divers’ actual experience levels.

  • Those data are not available at the moment.

Figures and Tables:

Table 1.  The bottom portion of the table is not clear.  Can this be more fully explained, and all of the abbreviations defined (should the J’s across the top be D’s for day?)?

  • Corrected

Figure 1.  There are word spacing issues within the text boxes.

  • Corrected